

# Perturbative and nonperturbative studies of CFTs with MN global symmetry

Johan Henriksson[1,2,3] and Andreas Stergiou[4]

**1** Dipartimento di Fisica E. Fermi, Università di Pisa, and INFN, Sezione di Pisa,
Largo Bruno Pontecorvo 3, 56127 Pisa, Italy
**2** Lincoln College, University of Oxford, Turl Street, Oxford, OX1 3DR, UK
**3** Mathematical Institute, University of Oxford, Woodstock Road, Oxford, OX2 6GG, UK
**4** Theoretical Division, MS B285, Los Alamos National Laboratory,
Los Alamos, NM 87545, USA

## Abstract

Fixed points in three dimensions described by conformal field theories with $MN_{m,n} = O(m)^n \rtimes S_n$ global symmetry have extensive applications in critical phenomena. Associated experimental data for $m = n = 2$ suggest the existence of two non-trivial fixed points, while the $\varepsilon$ expansion predicts only one, resulting in a puzzling state of affairs. A recent numerical conformal bootstrap study has found two kinks for small values of the parameters $m$ and $n$, with critical exponents in good agreement with experimental determinations in the $m = n = 2$ case. In this paper we investigate the fate of the corresponding fixed points as we vary the parameters $m$ and $n$. We find that one family of kinks approaches a perturbative limit as $m$ increases, and using large spin perturbation theory we construct a large $m$ expansion that fits well with the numerical data. This new expansion, akin to the large $N$ expansion of critical $O(N)$ models, is compatible with the fixed point found in the $\varepsilon$ expansion. For the other family of kinks, we find that it persists only for $n = 2$, where for large $m$ it approaches a non-perturbative limit with $\Delta_\phi \approx 0.75$. We investigate the spectrum in the case $MN_{100,2}$ and find consistency with expectations from the lightcone bootstrap.



# 1   Introduction

Second-order phase transitions display scale-invariant physics and are widely believed to be described by conformal field theories (CFTs), which arise at fixed points of the renormalization group (RG) flow. Due to universality, the physics at these phase transitions is independent of the underlying microscopic degrees of freedom, which means that the same CFT may describe a variety of systems. Many important applications of three-dimensional CFTs arise for non-zero temperature phase transitions, for instance critical liquid-vapor transitions, transitions between magnetic phases and structural phase transitions.

The observables of a conformal field theory, such as the critical exponents, can be extracted from the CFT data, which are the scaling dimensions and structure constants (OPE coefficients) of the local operators in the theory. One principal goal in the theory of critical phenomena is therefore to make precise determinations of the CFT data. A useful tool in studying CFTs relevant for three-dimensional systems is the Landau–Ginzburg–Wilson description, in which one writes down a quantum field theory with quartic interactions preserving a given global symmetry group. By tuning mass parameters (equivalent to tuning the temperature in experiments), the field theory flows under the RG to a fixed point preserving the same (or larger) global symmetry. Methods within this paradigm, such as the $\varepsilon$ expansion [1], produce in many cases values of the critical exponents that match well with experiments; see [2] for an extensive review.

Despite the remarkable success of the mentioned paradigm for many systems, including those with emergent $O(N)$ symmetry, one cannot rule out the existence of additional CFTs, not captured by the Landau–Ginzburg–Wilson description but still relevant for experimental realizations. An interesting case is systems with global symmetry group $MN_{m,n} = O(m)^n \rtimes S_n$. The most general Lagrangian that preserves $MN_{m,n}$ symmetry is

$$\mathscr{L} = \tfrac{1}{2}\partial_\mu \phi_i \partial^\mu \phi_i + \tfrac{1}{8}\lambda(\phi^2)^2 + \tfrac{1}{24}g\left[(\phi_1^2 + \cdots + \phi_m^2)^2 + \cdots + (\phi_{m(n-1)+1}^2 + \cdots + \phi_{mn}^2)^2\right], \quad (1)$$

where $\phi_i$ is an $mn$-dimensional vector under $MN_{m,n}$, $\phi^2 = \phi_i \phi_i$, and we keep only the kinetic term and the quartic interaction terms. For the two-coupling theory (1), the $\varepsilon$ expansion predicts only one fully-interacting fixed point with this global symmetry.[1] For the experimentally accessible case of $m = n = 2$, when the $O(2)^2 \rtimes S_2$ theory in (1) is equivalent to the more commonly discussed $O(2)^2/\mathbb{Z}_2$ theory [3], the critical exponents derived from this fixed point have not been successful in matching those measured in experiments with helimagnets and XY stacked triangular antiferromagnets, which cluster in two distinct regions; see Table 1. Additionally, the fixed point in the $\varepsilon$ expansion appears to have $g < 0$, which is inconsistent

---

[1]In addition to this fixed point, one finds also the free theory with $g = \lambda = 0$, the $O(mn)$ symmetric theory with $g = 0$ and $n$ decoupled $O(m)$ models with $\lambda = 0$.

Table 1: Experimental results for phase transitions described by $MN_{2,2}$ theory. These values are compiled from [2,4] and references therein.

| $MN_{2,2}$ | $\beta$ | $\nu$ |
|---|---|---|
| XY STAs | 0.24(2) | 0.55(5) |
| Tb | 0.23(4) | 0.53(4) |
| Ho, Dy | 0.39(4) | 0.57(4) |
| $NbO_2$ | $0.40^{+0.04}_{-0.07}$ | |

with the expected chiral universality class that should describe phase transitions in these systems [2,5]. [2] Evidence for the existence of further non-perturbative fixed points in the chiral region has been offered [6–8], but this has been disputed by other authors [4,9–14].

This contradictory set of observations motivated the recent study of MN symmetric theories using the non-perturbative (numerical) conformal bootstrap [15]. This method, proposed in [16] and extensively reviewed in [17], makes no assumptions on the underlying microscopic description and studies CFTs based only on global and conformal symmetry, unitarity, and consistency with the operator algebra (crossing symmetry). In agreement with the experimental

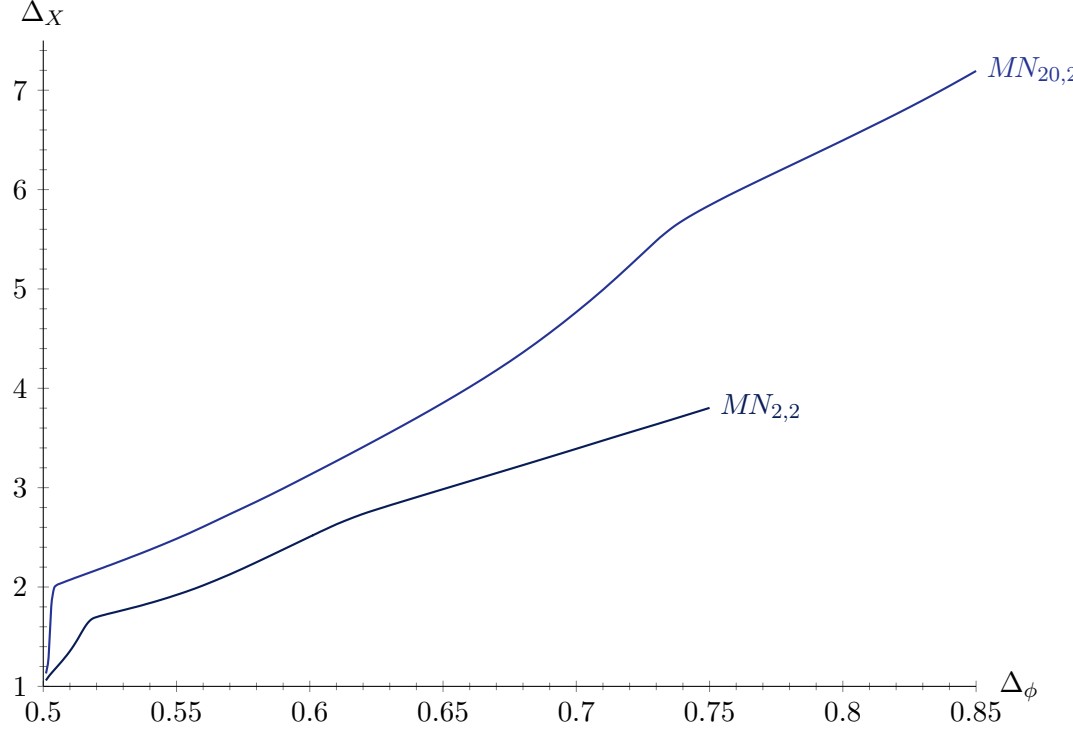

Figure 1: Bounds in the $(\Delta_\phi, \Delta_X)$ plane for 3D CFTs with $MN_{m,n}$ global symmetry with $(m,n) = (2,2)$ and $(m,n) = (20,2)$. The allowed region lies below the curves for the corresponding parameter values. These bounds are obtained with the use of PyCFTBoot [18] with parameters n_max=9, m_max=6, k_max=36 and l_max=26.

data, the conformal bootstrap study in [15] found evidence for the existence of two distinct CFTs with $MN_{2,2}$ symmetry, as can be seen from Fig. 1. This figure displays bounds on operator dimensions in the $(\Delta_\phi, \Delta_X)$ plane, where $\phi$ and $X$ respectively denote the smallest dimension

---

[2]In much of the literature, e.g. [2,5], a coupling $\nu$ is used instead of our $g$ in (1). The coupling $\nu$ and our coupling $g$ have opposite signs, so the chiral region, defined by $\nu > 0$ in the literature, corresponds to $g < 0$ in our notation.

operators transforming in the vector and a certain rank-two representation of the MN symmetry group. The region below the curves is the allowed parameter space in the respective theories. In various applications of the conformal bootstrap, it has been observed that a kink in the boundary of the allowed region is related to the existence of a conformal field theory with parameters near the location of the kink, and from the two kinks of the $MN_{2,2}$ curve in Fig. 1 the following values for the critical exponents were derived [15]:

$$\text{kink 1: } \beta = 0.293(3), \ \nu = 0.566(6), \qquad \text{kink 2: } \beta = 0.355(5), \ \nu = 0.576(8). \qquad (2)$$

While the critical exponents corresponding to the second kink show reasonable agreement, within uncertainties, with the results of [8,19], neither set of values is compatible with the predictions from the $\varepsilon$ expansion, which gives $\beta = 0.370(5)$ and $\nu = 0.715(10)$ [20,21].[3] Although it was speculated in [15] that the first kink may be related to the $\varepsilon$ expansion through the large $m$ limit, the results of that study were not sufficient to make any conclusive statements.

The purpose of the present paper is to make a more systematic study of the hypothetical CFTs living at the two kinks, by investigating the behavior of numerical bounds for various $(m, n)$. More specifically, we perform numerical conformal bootstrap studies for varying values of $m$, keeping $n$ fixed (for most of the work we keep $n = 2$, but we also obtain a bound for $m = 100$ and $n = 3, 4$). For $n = 2$ we find that, as we increase $m$, both kinks continue to exist, and for large $m$ the bound in the $(\Delta_\phi, \Delta_X)$ plane attain a profile similar to the case $MN_{20,2}$ displayed in Fig. 1. The large $m$ behavior of the CFT at each kink is subsequently studied.

First, we focus on the first kink, which approaches the value $(\Delta_\phi, \Delta_X) = (0.5, 2)$ as $m \to \infty$. In [15], it was observed that this limit is compatible with the results in the $\varepsilon$ expansion [3,21,24,25] expanded at large $m$, where indeed $\Delta_\phi \to \frac{d-2}{2}$ and $\Delta_X \to 2$ up to $m^{-1}$ corrections. We show here that the $m^{-1}$ corrections can be computed in a perturbative expansion similar to the usual large $N$ expansion of the $O(N)$ model (see e.g. [26]), where now the operator $X$ acts as the Hubbard–Stratonovich auxiliary field. Specifically, we use the analytic conformal bootstrap method of large spin perturbation theory, developed in [27–30], to compute $m^{-1}$ corrections to scaling dimensions and OPE coefficients. This expansion is valid for all spacetime dimensions $d \in (2, 4)$. Near $d = 4$, the results agree with the $\varepsilon$ expansion and in $d = 3$ we get good agreement with the non-perturbative bootstrap results for the first kink. This gives substantial evidence that we should view the first kink as describing a perturbative CFT with MN symmetry, existing for a range of $(m, n)$ and $d$ and connected to the $\varepsilon$ expansion via the large $m$ limit.

Second, we study the second kink for $n = 2$ and increasing $m$. The bounds in Fig. 1 reveal that, as we increase $m$, the values of the scaling dimensions $\Delta_\phi$ and $\Delta_X$ corresponding to this kink move far away from the free theory values. We perform a single correlator numerical bootstrap study where we search for bounds in the $(\Delta_\phi, \Delta_X)$ plane for increasing values of $m$, with the hope of finding a limit point at infinite $m$ for the position of the kink. The results show that the second kink continues to exist for all values of $m$ studied, and that the position in the $\Delta_\phi$ direction appears to stabilize near the value 0.75. The position in the $\Delta_X$ direction takes a value $\Delta_X \gtrsim 6$, but is highly sensitive to the numerical precision of the computation (number of derivatives in the functional used for the numerical bootstrap computations). These values for $\Delta_\phi$ and $\Delta_X$ show that, if there is a CFT corresponding to the second kink, it must be of a non-perturbative type.

To investigate further the potential CFT corresponding to the second kink, we focus on the case $MN_{100,2}$ and increase the numerical precision. We use the extremal functional method,

---

[3]The corresponding fixed point is called "complex cubic" in [20,21]. Regarding RG stability of this fixed point relative to the decoupled $O(2)$ one, we note that since the scaling dimension of the first singlet of the $O(2)$ model is slightly above 1.5 [22], this means that the decoupled $O(2)$ theory is stable in $d = 3$ [23].

developed in [31], to extract information about the spectrum of operators in the $\phi \times \phi$ OPE. The results give some hints of an organization of the leading twist operators in twist families, as must be the case according to the lightcone bootstrap [32–34]. For fixed $m = 100$ we also study $MN_{100,n}$ for $n = 3$ and $n = 4$, but for these values we find no second kink in the $(\Delta_\phi, \Delta_X)$ bound.

This paper is organized as follows. In section 2 we explain how to study MN symmetric CFTs in the bootstrap approach using crossing symmetry and unitarity. We introduce relevant notation and review the known perturbative results of the $\varepsilon$ expansion. In section 3 we show that the CFT corresponding to the first kink can be matched with a perturbative large $m$ expansion, which we construct using large spin perturbation theory. Further, we comment on the connection to the $\varepsilon$ expansion. In section 4 we use the non-perturbative numerical bootstrap to study the second kink, and discuss the twist families of the spectrum for the representative case $MN_{100,2}$. We finish with a discussion, and include some explicit results in an appendix.

For our numerical computations we have used PyCFTBoot [18], qboot [35] and SDPB [36].

## 2 Review

In this section we briefly review the constraints from unitarity and crossing symmetry on conformal field theories, with emphasis on theories with global MN symmetry. We then summarize the results from previous studies in the $\varepsilon$ expansion.

### 2.1 Unitarity and crossing in the presence of $MN_{m,n}$ symmetry

We consider the four-point correlator of $\phi^i$, $i = 1, \ldots, mn$, transforming in the vector representation $V$ of the $MN_{m,n} = O(m)^n \rtimes S_n$ global symmetry. More precisely, $O(m)$ acts by rotating the fields within each of the $n$ groups of $m$ fields, and $S_n$ permutes these groups. In terms of the conformal cross-ratios $u = \frac{x_{12}^2 x_{34}^2}{x_{13}^2 x_{24}^2}$ and $v = \frac{x_{14}^2 x_{23}^2}{x_{13}^2 x_{24}^2}$, with $x_{ij} = |x_i - x_j|$, the correlator takes the form

$$\langle \phi^i(x_1)\phi^j(x_2)\phi^k(x_3)\phi^l(x_4)\rangle = \frac{1}{x_{12}^{2\Delta_\phi} x_{34}^{2\Delta_\phi}} \sum_{R=S,X,Y,Z,A,B} \mathbf{T}_R^{ijkl} \mathcal{G}_R(u,v), \tag{3}$$

where $\mathbf{T}_R$ are projection tensors for the representations $R$ in the tensor product $V \otimes V = S \oplus X \oplus Y \oplus Z \oplus A \oplus B$. $S$ denotes the singlet representation with $\mathbf{T}_S^{ijkl} = \frac{1}{mn}\delta^{ij}\delta^{kl}$, while the remaining symbols denote rank-two symmetric ($X$, $Y$ and $Z$) and antisymmetric ($A$ and $B$) representations. We will not need the precise form of these projection tensors, which can be found in [15]. Each function $\mathcal{G}_R(u,v)$ admits a decomposition in conformal blocks,

$$\mathcal{G}_R(u,v) = \sum_{\mathcal{O} \in R} (-1)^{\ell_\mathcal{O}} \lambda_\mathcal{O}^2 g_{\Delta_\mathcal{O}, \ell_\mathcal{O}}(u,v), \tag{4}$$

where the sum runs over conformal primary operators of dimension $\Delta_\mathcal{O}$ and spin $\ell_\mathcal{O}$ transforming in the representation $R$. The conformal blocks, denoted by $g_{\Delta,\ell}(u,v)$, sum up the contribution to the correlator of a given primary and all its descendants, and are functions depending only on the cross-ratios and the dimension and spin of the primary.

We will adopt a notation where we denote by $R$, $R'$, $R''$ etc. the smallest dimension scalars in the representation $R$, and likewise by $R_\ell$, $R'_\ell$, $R''_\ell$ etc. the smallest dimension spin $\ell$ operators in the representation $R$. In any unitary CFT, the $S$ representation will contain the identity operator $\mathbb{1}$ with $\Delta = \ell = 0$,[4] and the stress-energy tensor $T = S_2$ with $\Delta_T = d$. In the

---

[4]This means we will denote by $S$ the smallest dimension operator different from the identity.

presence of a continuous global symmetry, a CFT contains in addition a conserved Noether current $J^\mu$ with $\Delta_J = d - 1$, which in our case resides in the $A$ representation: $J = A_1$.

Unitarity imposes constraints on the decomposition (4). Reality of the three-point functions $\langle \phi(x_1)\phi(x_2)\mathcal{O}(x_3) \rangle$ implies positivity of the expansion coefficients $\lambda_{\mathcal{O}}^2$, and positivity of two-point functions of descendants implies unitarity bounds, namely

$$\Delta_R \geqslant \tfrac{1}{2}(d-2), \qquad \Delta_{R_\ell} \geqslant d - 2 + \ell, \tag{5}$$

where the inequalities are saturated only for a free scalar and a conserved current respectively. Moreover, a non-trivial consequence of unitarity is Nachtmann's theorem [37], which states that the twists of the leading singlet operators, $\tau_{S,\ell} = \Delta_{S_\ell} - \ell$, form an upward convex function for all spin $\ell$ above some $\ell_0$, see [38] for a recent discussion.

Crossing symmetry follows from the invariance of the correlator (3) under exchanging pairs of insertion points. The invariance under $x_1 \leftrightarrow x_2$ is satisfied by each conformal block together with the fact that only operators of even/odd spins appear in each representation, while the invariance under $x_1 \leftrightarrow x_3$ leads to a non-trivial crossing equation which we will use. In the presence of global $MN_{m,n}$ symmetry, the crossing equation takes the form

$$\mathcal{G}_R(u,v) = \left(\frac{u}{v}\right)^{\Delta_\phi} \sum_{\widetilde{R}} M_{R\widetilde{R}}\, \mathcal{G}_{\widetilde{R}}(v,u), \tag{6}$$

where the crossing matrix $M_{R\widetilde{R}}$ in the basis $\{S, X, Y, Z, A, B\}$ is given by

$$M_{RR'} = \begin{pmatrix} \frac{1}{mn} & \frac{1}{mn} & \frac{1}{m} & 1 & -1 & -1 \\ \frac{n-1}{mn} & \frac{n-1}{mn} & \frac{n-1}{m} & -1 & 1-n & 1 \\ \frac{(m-1)(m+2)}{2mn} & \frac{(m-1)(m+2)}{2mn} & \frac{m-2}{2m} & 0 & \frac{m+2}{2} & 0 \\ \frac{n-1}{2n} & -\frac{1}{2n} & 0 & \frac{1}{2} & 0 & \frac{1}{2} \\ -\frac{m-1}{2mn} & -\frac{m-1}{2mn} & \frac{1}{2m} & 0 & \frac{1}{2} & 0 \\ -\frac{n-1}{2n} & \frac{1}{2n} & 0 & \frac{1}{2} & 0 & \frac{1}{2} \end{pmatrix}. \tag{7}$$

We refer to the left-hand side of (6) as the direct channel, and to the right-hand side as the crossed channel. In the analytic bootstrap approach in section 3.1, the crossing equation is expanded in the double lightcone limit $u \ll v \ll 1$, and the operators in the crossed channel will source corrections to the CFT data of the operators in the direct channel.

In the numerical bootstrap approach in section 4.1, the crossing equation is re-written in a form that treats the channels symmetrically. This form is given explicitly in [15], and the technical details can be found in that paper. The principles are the standard ones of the numerical conformal bootstrap [16, 17]: by acting on the crossing equation with a family of functionals, positivity of the squared OPE coefficients $\lambda_{\mathcal{O}}^2$ is turned into rigorous inequalities which rule out large regions of the space of allowed operator dimensions. For the theory at hand, we identify the potential fixed points by observing kinks in the bound in the $(\Delta_\phi, \Delta_X)$ plane, following [15]. To gain more information about the CFT at the position of the kink, we apply the extremal functional method developed in [31], which uses the fact that a functional in the vicinity of the CFT should vanish when applied to the conformal blocks of the operators present in the spectrum.

## 2.2 Results from previous studies in the $\varepsilon$ expansion

Conformal field theories with MN symmetry have been studied in the $d = 4 - \varepsilon$ expansion over a long time, [20, 21, 24, 25, 39, 40], most recently in section 5.2.2. of [3]. From the beta functions of the couplings $\lambda$ and $g$ in the Lagrangian (1), four fixed points are found in the $\varepsilon$ expansion: $mn$ free fields, $n$ decoupled critical $O(m)$ models, the critical $O(mn)$ model, and

the perturbative MN CFT. We focus on the MN CFT, which has $\lambda, g \neq 0$. At this fixed point, the scaling dimensions of the leading scalar operators have an expansion of the form

$$\Delta_\phi = 1 - \frac{\varepsilon}{2} + \frac{m(n-1)[(m+2)mn - 10m + 16]}{4 C_{mn}^2}\varepsilon^2 + \gamma_\phi^{(3)}\varepsilon^3 + O(\varepsilon^4), \tag{8}$$

$$\Delta_S = 2 - \varepsilon + \frac{6m(n-1)}{C_{mn}}\varepsilon + \gamma_S^{(2)}\varepsilon^2 + \gamma_S^{(3)}\varepsilon^3 + +O(\varepsilon^4), \tag{9}$$

$$\Delta_X = 2 - \varepsilon + \frac{m((m+2)n - 6)}{C_{mn}}\varepsilon + \gamma_X^{(2)}\varepsilon^2 + \gamma_X^{(3)}\varepsilon^3 + O(\varepsilon^4), \tag{10}$$

$$\Delta_Y = 2 - \varepsilon + \frac{2m(n-1)}{C_{mn}}\varepsilon + \gamma_Y^{(2)}\varepsilon^2 + \gamma_Y^{(3)}\varepsilon^3 + O(\varepsilon^4), \tag{11}$$

$$\Delta_Z = 2 - \varepsilon - \frac{2(m-4)}{C_{mn}}\varepsilon + \gamma_Z^{(2)}\varepsilon^2 + \gamma_Z^{(3)}\varepsilon^3 + O(\varepsilon^4), \tag{12}$$

where $C_{mn} = (m+8)mn - 16(m-1)$. In the above expressions we have omitted the explicit form of the order $\varepsilon^2$ and $\varepsilon^3$ corrections, which were derived in [3][5] and are available by an email request to the authors. Moreover, the eigenvalues of the stability matrix are [3]

$$\omega_1 = \varepsilon + \omega_1^{(2)}\varepsilon^2 + \omega_1^{(3)}\varepsilon^3 + O(\varepsilon^4), \tag{13}$$

$$\omega_2 = -\frac{(m-4)(mn-4)}{C_{mn}}\varepsilon + \omega_2^{(2)}\varepsilon^2 + \omega_2^{(3)}\varepsilon^3 + O(\varepsilon^4), \tag{14}$$

which correspond to $\omega_i = \Delta_i - d$ for the singlet operators of $\phi^4$ type in the theory.

For the specific case of $m = 2$, the order $\varepsilon^4$ renormalization was performed in [20, 21], giving $\Delta_\phi$, $\Delta_S$, $\omega_1$ and $\omega_2$ to this order. For the physically relevant cases $n = 2$ and $n = 3$, a Borel–Leroy resummation was performed to give estimates for the critical exponents $\gamma$, $\nu$, $\eta$ in three dimensions; see section 3.2 below.

## 3 The perturbative fixed point at large $m$

In this section we will derive a large $m$ expansion for $MN_{m,n}$ symmetric CFTs, and show that it gives predictions that match well with those found in the numerical bootstrap for the first kink. The existence of this expansion establishes the perturbative nature of the corresponding family of CFTs .

### 3.1 Analytic expansion from large spin perturbation theory

Expanding the expressions (9)–(12) for the scalar operators at large $m$ we observe that

$$\Delta_X = 2 + O(m^{-1}), \tag{15}$$

whereas the scalar operators in the $S$, $Y$ and $Z$ representations all satisfy $\Delta = 2 - \varepsilon + O(m^{-1})$. This observation indicates that there exists, for all $d \in (2,4]$, a large $m$ expansion where $\Delta_X = 2 + O(m^{-1})$, and $\Delta_R = d - 2 + O(m^{-1})$ for $R = S, Y, Z$. These values are consistent with a description in terms of Hubbard–Stratonovich auxiliary fields, similar to the large $N$ expansion of the critical $O(N)$ model.

In [30], based on [28, 29], it was described how to use large spin perturbation theory to extract properties of $\phi^4$ theories with Hubbard–Stratonovich auxiliary fields. We will follow

---

[5]The leading $m$ dependence is given in equation (5.104) and (5.105) in the arXiv submission of [3], where $N = mn$ and $\sigma = S$, $\rho_1 = X$, $\rho_2 = Z$ and $\rho_3 = Y$.

this approach, which means that we assume that at large $m$ the operator spectrum is that of mean field theory for $\phi$ with $\Delta_\phi = \frac{d-2}{2} + O(m^{-1})$, but with the bilinear scalar in the $X$ representation replaced by a Hubbard–Stratonovich field $X$ of dimension $\Delta_X = 2 + O(m^{-1})$. The framework of [30] will then show what operator dimensions, in the large $m$ expansion, are consistent with these assumptions.

Let us briefly review the method of large spin perturbation theory, which in the present case will be a perturbation of mean field theory, i.e. we assume that each representation $R$ in the tensor product $V \otimes V$ contains operators of spin $\ell$ and scaling dimensions $2\Delta_\phi + 2k + \ell + O(m^{-1})$. The operators with $k > 0$ have OPE coefficients that are suppressed in the $1/m$ expansion,[6] and we can therefore focus on the leading twist operators, which we denote by $R_\ell$. These are bilinear operators of the schematic form $\phi \partial^\ell \phi$ and acquire individual anomalous dimensions

$$\Delta_{R_\ell} = 2\Delta_\phi + \ell + \gamma_{R_\ell}, \tag{16}$$

where $\gamma_{R,\ell}$ is of order $O(m^{-1})$. Symmetry under $x_1 \leftrightarrow x_2$ constrains the leading twist operators such that those in the $S, X, Y, Z$ representations have even spin, and those in the $A, B$ representations have odd spin.

In large spin perturbation theory, the OPE coefficients $\lambda^2_{\phi\phi R_\ell}$ and the anomalous dimensions $\gamma_{R_\ell}$ of spinning operators in the direct channel are computed using the Lorentzian inversion formula [42]. The integrand of the inversion formula is proportional to the double-discontinuity dDisc$[\mathcal{G}_R(u,v)]$, defined as the difference between the correlator and its two analytic continuations around $v = 0$. In the limit $u \ll v \ll 0$, and in an expansion in $m^{-1}$, the double-discontinuity can be computed from crossed-channel operators, i.e. those appearing in the conformal block decomposition of the right-hand side of (6). The contribution to the $R_\ell$ from an operator $\mathcal{O}$ in the $\tilde{R}$ representation is proportional to

$$\text{dDisc}|_{\mathcal{O}} \sim M_{R\tilde{R}} \lambda^2_{\phi\phi\mathcal{O}} \sin^2\left[\frac{\pi}{2}(\tau_\mathcal{O} - 2\Delta_\phi)\right], \tag{17}$$

where the argument of the squared sine is derived from the $v \to 0$ scaling of $v^{-\Delta_\phi} g_{\Delta_\mathcal{O}, \ell_\mathcal{O}}(v, u) \sim v^{\frac{1}{2}\tau_\mathcal{O} - \Delta_\phi}$ with $\tau_\mathcal{O} = \Delta_\mathcal{O} - \ell_\mathcal{O}$. The appearance of the $\sin^2$ factor means that the contribution from mean field theory operators will be suppressed by their squared anomalous dimension.

In order to apply the framework of [30], we assume that the operator $X$ has an OPE coefficient of the form

$$\lambda^2_{\phi\phi X} = \frac{a_X}{m} + O(m^{-2}), \tag{18}$$

for some constant $a_X$ depending on $n$ and spacetime dimension $d$. The contribution to the CFT data in the direct channel is then computed using the inversion formula from double-discontinuities (17) of crossed-channel operators. The identity operator $\mathbb{1}$ generates the leading OPE coefficients of $R_\ell$, and the operator $X$ gives a leading order contribution to $\gamma_{R_\ell}$ in all representations $R$. In the representations $Y, Z, A, B$ the crossed-channel operators $\mathbb{1}$ and $X$ provide the only contributions at order $m^{-1}$, and, using the formulas given in [30], we can write down the scaling dimensions[7]

$$\Delta_{Y_\ell} = \Delta_{A_\ell} = \ell + 2\Delta_\phi - \frac{a_X}{2(\ell + 1/2)(\ell - 1/2)m} + O(m^{-2}), \tag{19}$$

$$\Delta_{Z_\ell} = \Delta_{B_\ell} = \ell + 2\Delta_\phi + \frac{a_X}{2(\ell + 1/2)(\ell - 1/2)(n-1)m} + O(m^{-2}). \tag{20}$$

Recall that the spin $\ell$ is even for $Y$ and $Z$, and odd for $A$ and $B$.

---

[6] This follows immediately from the expression for the mean field theory OPE coefficients derived in [41], upon inserting $\Delta_\phi = \frac{d-2}{2} + O(m^{-1})$.

[7] We present only the values in $d = 3$ dimensions, results for generic $d$ are given in Appendix A.

Due to the combined $m$ dependence of all three factors in (17), the anomalous dimensions in the $S$ and $X$ representations will get leading order contributions from $X$ as well as from the $R_\ell$ in the other four representations. In the language of [30], we therefore have group I $= \{Y, Z, A, B\}$, and group II $= \{S, X\}$. Evaluating the formulas of that paper gives

$$\Delta_{S_\ell} = \ell + 2\Delta_\phi - \frac{a_X}{2(\ell+1/2)(\ell-1/2)m} - \frac{\pi^2 \ell\, a_X^2 n}{4(\ell+1/2)(\ell-1/2)(n-1)m} + O(m^{-2}), \quad (21)$$

$$\Delta_{X_\ell} = \ell + 2\Delta_\phi - \frac{a_X}{2(\ell+1/2)(\ell-1/2)m} - \frac{\pi^2 \ell\, a_X^2 n(n-2)}{4(\ell+1/2)(\ell-1/2)(n-1)^2 m} + O(m^{-2}). \quad (22)$$

The expressions (19)–(22) depend on two unknowns: the constant $a_X$ introduced in (18), and the leading anomalous dimension $\gamma_\phi^{(1)}$ defined by $\Delta_\phi = \frac{d-2}{2} + \gamma_\phi^{(1)}/m + O(m^{-2})$. However, conservation of the global symmetry current and the stress-energy tensor gives the two equations $\Delta_{A_1} = d-1$, $\Delta_{S_2} = d$. The latter equation is quadratic in $a_X$, and we get two solutions: $a_X = \gamma_\phi^{(1)} = 0$ and

$$a_X = \frac{4(n-1)}{\pi^2 n}, \qquad \gamma_\phi^{(1)} = \frac{4(n-1)}{3\pi^2 n}. \quad (23)$$

Choosing the non-trivial solution, we have fixed the leading order anomalous dimensions of all leading twist spinning operators in the theory.

We have also computed the corrections to the OPE coefficients of these operators, defined with respect to the leading order result given by $M_{RS}$ times the OPE coefficients of mean field theory. In general, these results are not particularly illuminating, but specifying to the conserved operators we extract the corrections to the central charge and the current central charge,[8]

$$\frac{C_T}{C_{T,\text{free}}} = 1 - \frac{40(n-1)}{9n} \frac{1}{\pi^2 m} + O(m^{-2}), \quad (24)$$

$$\frac{C_J}{C_{J,\text{free}}} = 1 - \frac{64(n-1)}{9n} \frac{1}{\pi^2 m} + O(m^{-2}). \quad (25)$$

The most important class of observables is the dimensions of the leading scalar operators in each representation. Unfortunately, spin zero is beyond the guaranteed region of convergence of the Lorentzian inversion formula and it is not *a priori* clear how to extract these values. However, in the large $N$ expansion of theories with $O(N)$ [29] and $O(m) \times O(N)$ [30] global symmetry, it has been observed that evaluating $\Delta_{R_\ell}$ for $\ell = 0$ correctly reproduces the dimensions of the scalar operators as computed by independent methods. If we assume that this is the case also for our MN symmetric theory, we can find the scalar operator dimensions by demanding that $\Delta_X = d - \Delta_{X_{\ell=0}}$ and that $\Delta_R = \Delta_{R_{\ell=0}}$ for $R = S, Y, Z$. Further support for this assumption is that the expressions derived from this assumptions, evaluated for $d = 4-\varepsilon$, agree with the expressions (9)–(12) in the overlap of the orders: $\varepsilon^3/m$. In three dimensions we find the following dimensions of the scalar operators:

$$\Delta_\phi = \frac{1}{2} + \frac{4(n-1)}{3n} \frac{1}{\pi^2 m} + O(m^{-2}), \quad (26)$$

$$\Delta_S = 1 + \frac{32(n-1)}{3n} \frac{1}{\pi^2 m} + O(m^{-2}), \quad (27)$$

$$\Delta_X = 2 - \frac{32(n-1)}{3n} \frac{1}{\pi^2 m} + O(m^{-2}), \quad (28)$$

---

[8]The observant reader may note that these results agree with those of one free and $n-1$ interacting $O(m)$ models. This agreement is broken at higher orders in $m^{-1}$, which can be seen from the $\varepsilon$ expansion, where $C_T/C_{T,\text{free}} = 1 - 5\gamma_\phi^{(2)} \varepsilon^2/3 + O(\varepsilon^3)$ [30].

$$\Delta_Y = 1 + \frac{32(n-1)}{3n} \frac{1}{\pi^2 m} + O(m^{-2}), \tag{29}$$

$$\Delta_Z = 1 + \frac{8(n-4)}{3n} \frac{1}{\pi^2 m} + O(m^{-2}), \tag{30}$$

and results for generic spacetime dimension $d$ are presented in Appendix A.

## 3.2 Comparison with numerical results

The results (26)–(30) can now be compared with the numerical bootstrap results for the first kink. In Fig. 2 we display the bounds in the $(\Delta_\phi, \Delta_X)$ plane from Fig. 2 of [15], together with our new large $m$ results (26), (28), as well as the $\varepsilon^3$ results (8), (10)[9] from the literature. We

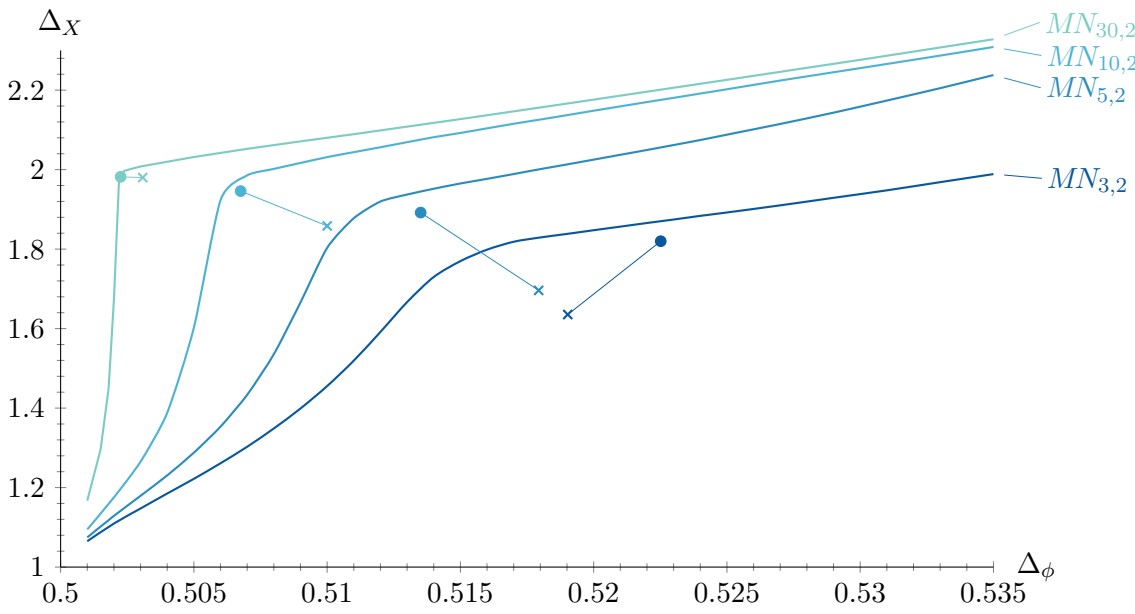

Figure 2: Bounds and corresponding locations of fixed points given as dots for large $m$ and crosses for $\varepsilon$ expansion results. The lines connecting dots and crosses are drawn to help illustrate results pertaining to the same theory.

see that the agreement is good between all three methods for $m \gg 1$, and that the large $m$ expansion better captures the finite $m$ behavior than does the $\varepsilon$ expansion.

Our new results in the large $m$ expansion can also be used to derive predictions for the critical exponents using the relations

$$\eta = 2 - d + 2\Delta_\phi, \qquad \nu^{-1} = d - \Delta_S, \qquad \alpha = 2 - \nu d, \tag{31}$$

$$\beta = \nu \Delta_\phi, \qquad \gamma = \nu(d - 2\Delta_\phi), \qquad \phi_\kappa = \nu(d - \Delta_Z). \tag{32}$$

For the physically relevant case $MN_{2,2}$, our new predictions for $\beta$ and $\nu$ from the large $m$ expansion are closer to numerical bootstrap results, as well as experiments and Monte Carlo results, than predictions from the $\varepsilon$ expansion, as can be seen in Table 2. Also for the chiral cross-over exponent $\phi_\kappa$, the value 1.2343 derived from our large $m$ results[10] compares favorably with the Monte Carlo value 1.22(6) [43].

---

[9]In producing this graph, as well as the corresponding values in Table 2, we have used the truncated results of the $\varepsilon$ expansion at order $\varepsilon^3$. We comment on this in section 3.3.

[10]We first estimated $\Delta_S$ and $\Delta_Z$ by evaluating the truncated expansions (27) and (30) for $m = n = 2$, before using (32) to find $\phi_\kappa$.

Table 2: Comparison of data for MN symmetric theories across various methods. The truncated series denote truncation to orders $\varepsilon^3$ and $m^{-1}$ respectively for the scaling dimensions. These numerical values are then used in (31) and (32) to give estimates for $\beta$ and $\nu$.

| $MN_{2,2}$ | $\Delta_\phi$ | $\Delta_S$ | $\Delta_X$ | $\beta$ | $\nu$ |
|---|---|---|---|---|---|
| XY STA | | | | 0.24(2) | 0.55(5) |
| Tb | | | | 0.23(4) | 0.53(4) |
| Monte Carlo [43] | | | | 0.253(10) | 0.54(2) |
| Monte Carlo [8] | | | | 0.317(35) | 0.63(7) |
| Numerical bootstrap [15] | 0.518(1) | 1.233(20) | 1.676(10) | 0.293(3) | 0.566(6) |
| $\varepsilon^4$ resummation [20, 21] | | | | 0.370(5) | 0.715(10) |
| $\varepsilon$ expansion (trunc.) | 0.5191 | 1.5052 | 1.3808 | 0.3473 | 0.6690 |
| Large $m$ expansion (trunc.) | 0.5338 | 1.2702 | 1.7298 | 0.3086 | 0.5781 |
| $MN_{2,3}$ | $\Delta_\phi$ | $\Delta_S$ | $\Delta_X$ | $\beta$ | $\nu$ |
| Numerical bootstrap [15] | 0.518(1) | 1.279(20) | 1.590(10) | 0.301(3) | 0.581(6) |
| $\varepsilon^4$ resummation [20, 21] | | | | 0.363(6) | 0.702(10) |
| $\varepsilon$ expansion (trunc.) | 0.5186 | 1.4642 | 1.3976 | 0.3377 | 0.6511 |
| Large $m$ expansion (trunc.) | 0.5450 | 1.3603 | 1.6398 | 0.3324 | 0.6099 |
| $MN_{20,2}$ | $\Delta_\phi$ | $\Delta_S$ | $\Delta_X$ | $\beta$ | $\nu$ |
| Numerical bootstrap | 0.5032(1) | 1.025(20) | 1.965(10) | 0.2548(26) | 0.506(5) |
| $\varepsilon$ expansion (trunc.) | 0.5047 | 1.0215 | 1.9606 | 0.2551 | 0.5054 |
| Large $m$ expansion (trunc.) | 0.5034 | 1.0270 | 1.9730 | 0.2551 | 0.5068 |

### 3.3 Connection to the $\varepsilon$ expansion

As we mentioned in the introduction, results derived in the $\varepsilon$ expansion have not been successful in matching the experimental values observed in the cases $MN_{2,2}$ and $MN_{2,3}$. This is in contrast to critical phenomena described by CFTs with several other symmetry groups, where the results in the $\varepsilon$ expansion give surprisingly good agreement with experimental data as well as non-perturbative results from Monte Carlo simulations and numerical conformal bootstrap.

The lack of agreement between bootstrap and $\varepsilon$ expansion results in the $MN_{2,2}$ case may be taken as a sign that the fixed point found of the $\varepsilon$ expansion, as discussed in section 2.2, is not connected to the CFT describing the critical phenomena in three dimensions. Our results strongly indicate the contrary, and that the connection is manifest through the large $m$ expansion derived above. Specifically, near four dimensions our new analytic results agree with the $\varepsilon$ expansion, and in three dimensions, the large $m$ expansion is connected to the finite $m$ CFTs through the family of kinks displayed in Fig. 2.

We note that for the larger values of $m$ there is good agreement between all three methods: large $m$ expansion, $\varepsilon$ expansion, and numerical conformal bootstrap. For the lower values of $m$, our new large $m$ expansion evaluates at a point closer to the corresponding kink than does the $\varepsilon$ expansion. For the latter, we have simply used direct truncation of the order $\varepsilon^3$ results; alternatively one could use Padé approximants or various resummation techniques.[11] In Table 2 we extended this comparison to more observables, and again we get an improved agreement with the numerical bootstrap compared to the $\varepsilon$ expansion. Note, for instance, that for small $m$ the $\varepsilon$ expansion predicts that $\Delta_X < \Delta_S$, which is inconsistent with the bootstrap

---

[11]We find that the Padé approximants constructed from the order $\varepsilon^3$ results contain spurious poles in the region $\varepsilon \leqslant 1$, and since the $\varepsilon$ expansion is not the focus of this paper we have not attempted any resummation methods. Note that the resummed $\varepsilon$ expansion of [21], included in Table 2, does not give any improvement compared to a direct truncation.

results.

In Table 2, for the $MN_{2,2}$ case, we have also included some results from experiments and Monte Carlo simulations, which in the literature are assigned to the chiral universality class. As mentioned in the introduction, the $MN_{2,2}$ fixed point obtained in the $\varepsilon$ expansion has $g < 0$ in (1), which means that it should not be applicable to these cases. However, $g < 0$ in the $\varepsilon$ expansion does not guarantee that $g < 0$ as $\varepsilon$ becomes finite, and therefore it cannot be ruled out that at $\varepsilon = 1$ the fixed point of kink 1 in fact has $g > 0$ and thus lies in the chiral region. Unfortunately, we are currently unable to probe the sign of $g$ using our bootstrap methods.

While the values for the critical exponent $\nu$ show good agreement across experiments, bootstrap, Monte Carlo and large $m$ expansion, the situation for the exponent $\beta$ is more concerning. In fact, as already pointed out in the literature [2,4], some of the experimental and Monte Carlo values do not satisfy the constraint $2\beta - \nu \geqslant 0$ as implied by the unitarity bound (5) for $\Delta_\phi$. This inconsistency could be explained by unknown systematic errors of these methods, or that they are not measuring the exponents at criticality. We do not wish to comment further on this, more than to point out that our results give exponents consistent with unitarity.

## 4    The non-perturbative fixed point at large $m$

In this section we study, using the numerical bootstrap, the large $m$ limit of the second kink for $MN_{m,n}$ symmetric theories. While two kinks are clearly visible for $MN_{2,3}$ in [15, Fig. 4], we find that the second kink only persists at large $m$ for the case $n = 2$, and we focus our attention to the cases $MN_{m,2}$ for various $m$.

### 4.1    Numerical bootstrap study

Our first set of results consists of bootstrap bounds for $MN_{m,2}$ theories for large values of $m$; see Fig. 3. These bounds show that the kink persists at large $m$ and that its position stabilizes close to $\Delta_\phi = 0.75$. However, the kink still moves significantly in the $\Delta_X$ direction.

The position for $\Delta_X$ of the kink is not stable upon increasing the number of derivatives; see Fig. 4. It appears, however, that $\Delta_\phi$ is fairly stable near the value 0.75. We conjecture that

$$\Delta_\phi|_{m\to\infty,n=2} = 0.75(1).  \tag{33}$$

For our strongest numerics we used `qboot` [35] with parameters `prec=1300`, `n_Max=560`, `lambda=51`, `numax=30` and the set of spins $\{0, \ldots, 80, 85, 86, 89, 90, 93, 94, 97, 98, 101, 102, 105, 106, 109, 110, 111, 112, 115, 116, 119, 120\}$.

We also obtained bounds for $m = 100$ with $n = 3, 4$. As we see in Fig. 5, the kink is clearly present only for $n = 2$. This suggests that there exists a critical line $n_c(m)$ below which we have two distinct CFTs with $MN_{m,n}$ global symmetry.

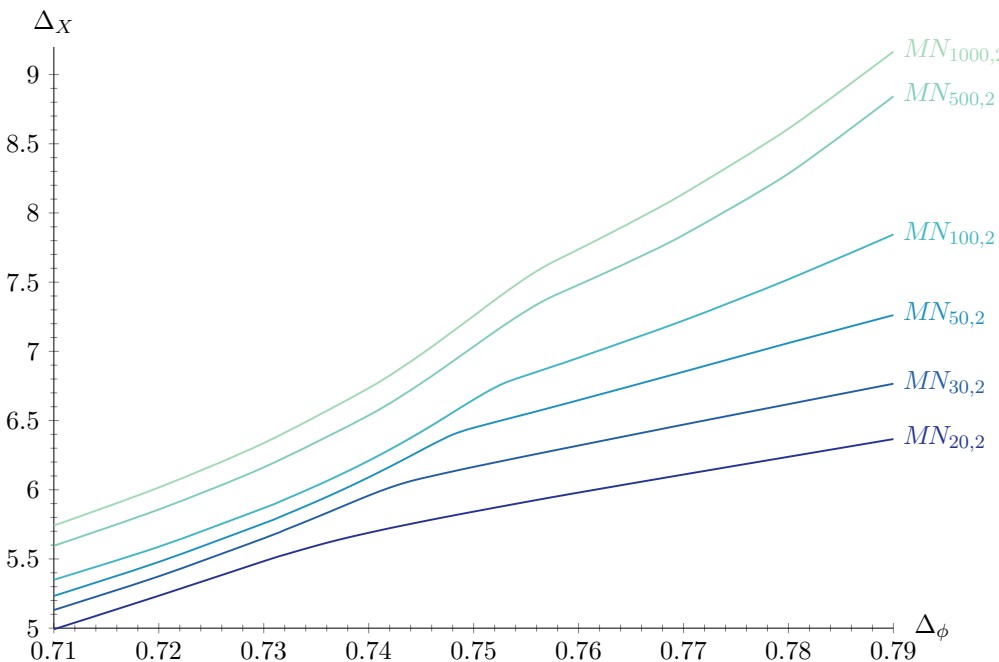

Figure 3: Bounds for 3D CFTs with $MN_{m,2}$ global symmetry for various values of $m$. The allowed region is below the curves in the corresponding theories. These bounds are obtained with the use of PyCFTBoot [18] with parameters n_max=9, m_max=6, k_max=36 and l_max=26.

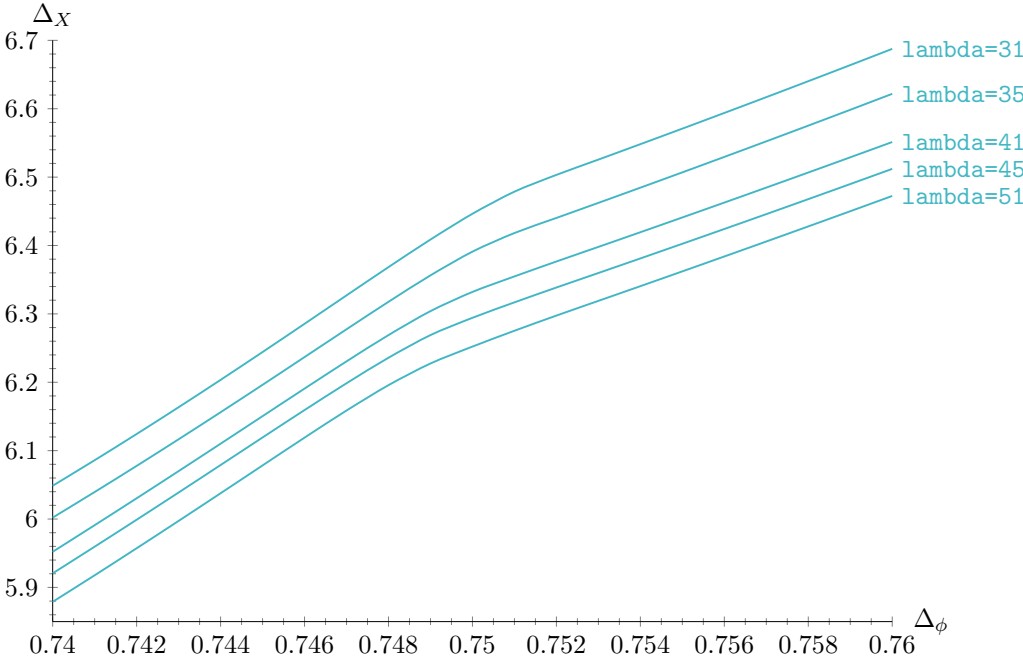

Figure 4: Bounds for 3D CFTs with $MN_{100,2}$ global symmetry with increasing numerical strength (top to bottom). For these bounds we used qboot [35]. These bounds are all stronger than the corresponding bounds in Fig. 3. The latter are obtained with qboot with the choice $\Lambda \approx 25$.

Subsequently, we focused on the values $(m, n) = (100, 2)$, and extracted the spectrum using

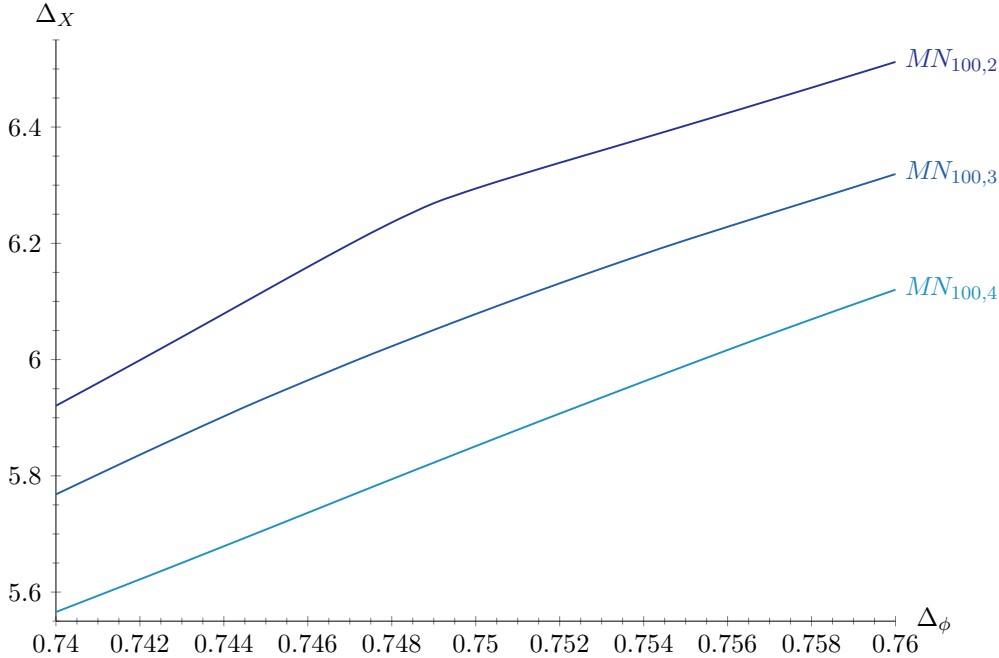

Figure 5: Bounds for 3D CFTs with $MN_{100,n}$ global symmetry for various values of $n$. The allowed region is below the curves in the corresponding theories. For these bounds we used qboot [35] with parameters prec=1200, n_Max=520, lambda=45, numax=26 and the set of spins $\{0, \ldots, 60, 63, 64, 66, 67, 73, 74, 77, 78, 81, 82, 85, 86, 89, 90, 93, 94, 97, 98\}$.

the extremal functional method [31]; see Fig. 6 and Fig. 7. We present results for $\Delta_\phi = 0.75$, for which we used PyCFTBoot [18] with parameters n_max=13, m_max=10, k_max=50 and l_max=40. From Fig. 4 we we estimate the scalar operator dimensions, for $(m, n) = (100, 2)$, to

$$\Delta_\phi = 0.75(1), \tag{34}$$

$$\Delta_X = 6.1(4). \tag{35}$$

From Figs. 6 and 7 we then read off the dimensions of the leading scalar operators to

$$\Delta_S \approx 1.35, \tag{36}$$

$$\Delta_Y \approx 0.8, \tag{37}$$

$$\Delta_Z \approx 0.6. \tag{38}$$

It is interesting to note that these results have $\Delta_Z < \Delta_\phi$. The small values for $\Delta_Y$ and $\Delta_Z$ suggest that a mixed correlator bootstrap involving the operators $Y$ and/or $Z$ may give results that are quite constraining.

## 4.2 Nonperturbative aspects of the large $m$ theory

The results from our numerical bootstrap show that the second kink continues to exist for all values of $m \geqslant 2$, indicating the existence of a corresponding CFT, in the sense of a set of conformal primary operators with scaling dimensions and OPE coefficients consistent with unitarity and crossing. We have only considered the constraints from the $\langle \phi\phi\phi\phi \rangle$ correlator, and numerical studies using a multi-correlator approach will either give further constraints on the candidate CFT or disprove its existence.



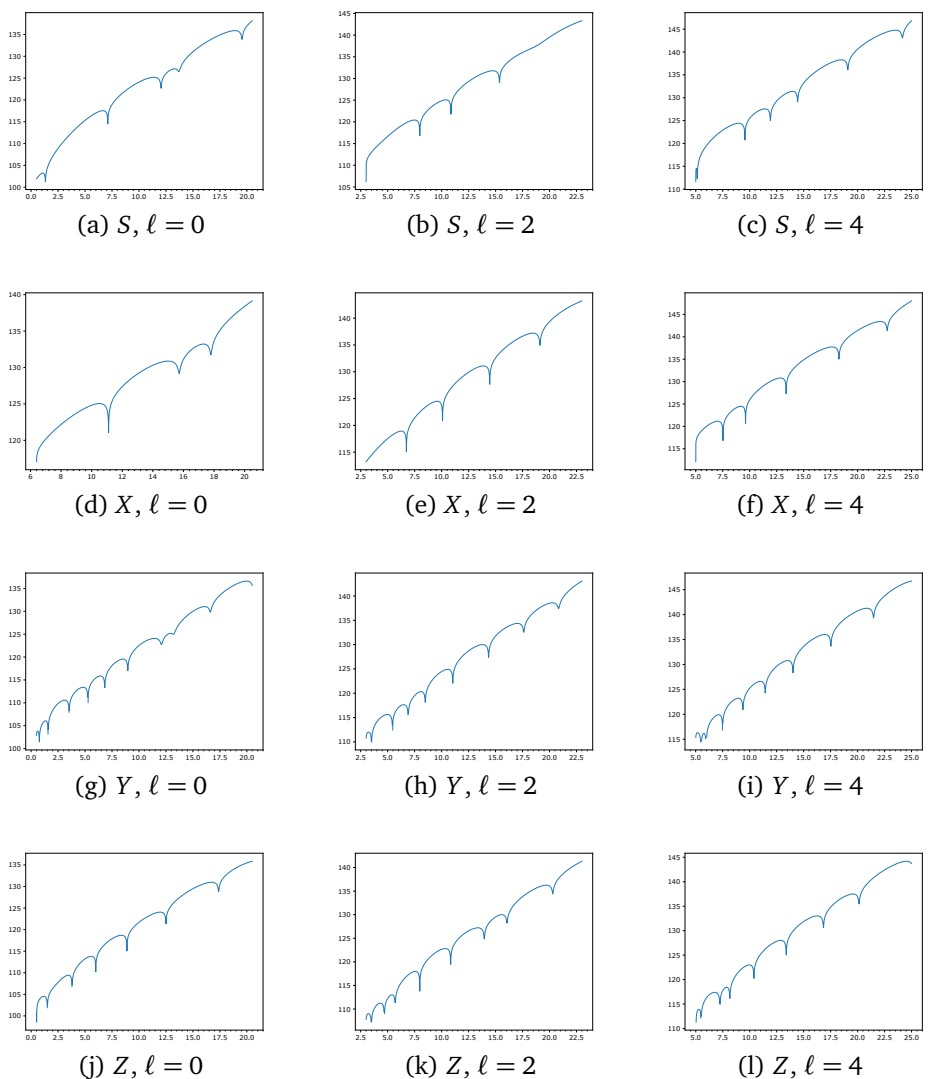

Figure 6: Plots from the extremal functional method for the even representations. In the horizontal axis we plot the scaling dimension and in the vertical the logarithm of the action of the functional on convolved conformal blocks (denoted by $F_{\Delta,\ell}$ in [31]).

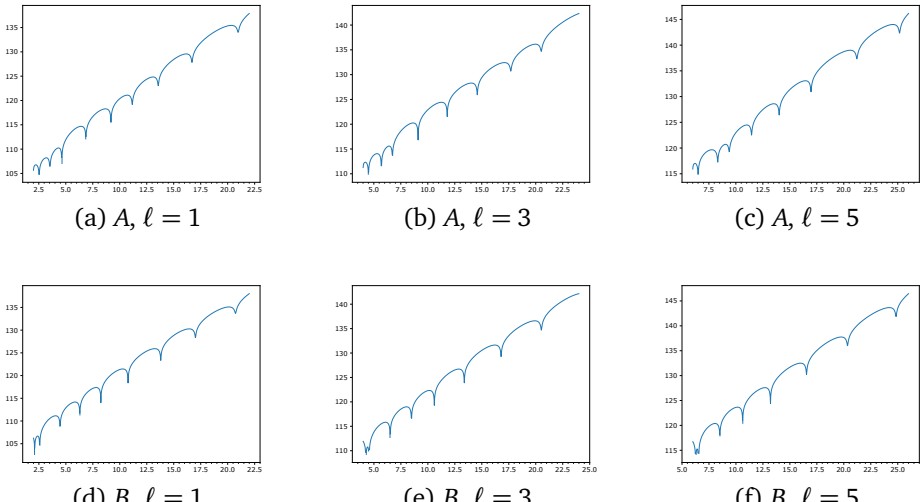

(a) $A, \ell = 1$      (b) $A, \ell = 3$      (c) $A, \ell = 5$

(d) $B, \ell = 1$      (e) $B, \ell = 3$      (f) $B, \ell = 5$

Figure 7: Plots from the extremal functional method for the odd representations.

The motivation for our work was to see if the candidate theory approaches a simplifying limit as $m \to \infty$, for instance a perturbation from a theory of generalized free fields. If this were the case, it could potentially be studied using perturbative methods just like we did for the first kink in section 3. Our results show that this is not the case, meaning that the candidate CFT remains non-perturbative, or "strongly coupled" for all values of $m$. It will therefore be challenging to corroborate the existence of this CFT.

There is, however, one test that any conformal field theory must pass, be it perturbative or not, namely consistency with the predictions from the lightcone bootstrap [32, 33]. These papers proved the *twist additivity*, stating that any conformal field theory containing operators $\mathcal{O}_1$ and $\mathcal{O}_2$, must also contain an infinite family of spinning operators $[\mathcal{O}_1, \mathcal{O}_2]_{0,\ell}$ [12] with twists (recall that $\tau = \Delta - \ell$)

$$\lim_{\ell \to \infty} \tau_\ell = \tau_\infty = \tau_1 + \tau_2. \tag{39}$$

This statement means that a given theory will contain many different accumulation points $\tau_\infty$, see [44] for observations in the $\varepsilon$ expansion of the $O(N)$ model predating the lightcone bootstrap, however not all these values may be visible in a given correlator. In fact, the only accumulation point guaranteed to exist in the direct channel of the $\langle \mathcal{O}_1 \mathcal{O}_2 \mathcal{O}_2 \mathcal{O}_1 \rangle$ is $\tau_1 + \tau_2$, but other values are not excluded. The conclusion is that we expect that in our candidate CFT the value

$$\tau_\infty = 2\Delta_\phi = 1.50(2) \tag{40}$$

is a twist accumulation point, and operators in the twist family corresponding to this value should be visible in our numerical bootstrap study.

In a weakly coupled theory, where $\Delta_\phi$ is close to the scalar unitarity bound $\frac{d-2}{2}$, the twists of the double-twist operators $[\phi, \phi]_{0,\ell}$ are also close to the spinning unitarity bound $d - 2$, c.f. (5). In such a theory, the double-twist operators have leading twist, and will include the stress-energy tensor at $\ell = 2$.

In the generic case, where $\Delta_\phi - \frac{d-2}{2}$ is finite, there are three possibilities:

- The double-twist operators $[\phi, \phi]_{0,\ell}$ remain the operators at leading twist for each spin, which means that they acquire "large" anomalous dimensions $\gamma_\ell = \tau_\ell - \tau_\infty$ in order to accommodate the stress-energy tensor at $\ell = 2$. This is the situation in 3D critical $O(N)$

---

[12]The interacting theory also contains subleading twist families $[\mathcal{O}_1, \mathcal{O}_2]_{k,\ell}$ with twists approaching $\tau_1 + \tau_2 + 2k$.

models at finite $N$, and indeed also in the perturbative fixed point of section 3 in this paper.[13]

- The stress-energy tensor belongs to an additional twist family below the double-twist operators. This case happens for instance when $\phi$ is a composite operator in weakly coupled theories. An example is correlators of gauge-invariant operators in weakly coupled $\mathcal{N} = 4$ SYM, where the double-twist family consists of double-trace operators and the additional leading twist family consists of the single-trace weakly broken currents.

- The stress-energy tensor and other conserved currents are isolated operators not belonging to any twist family. This happens in $\mathcal{N} = 4$ in the strong coupling expansion, where the lower limit of analyticity in spin is shifted upwards [42], but is not expected for a *bona fide* CFT with usual Regge behavior.

We now study the plots in Figs. 6 and 7 to see if they are consistent with any of the mentioned scenarios. Since the position of the kink could not be precisely determined, we do not expect these plots to give very precise values for the operator dimensions. We can, however, look at the qualitative behaviour of the lowest dimension operators.

In the $S$ and $X$ representations, the spectrum plots are comparatively sparse. There the lowest dimension operators at each spin are consistent with the first scenario, with large anomalous dimensions. In the singlet representation, these anomalous dimensions are negative, which is consistent with Nachtmann's theorem (convexity) and with the stress-energy tensor appearing at $\ell = 2$. In the $X$ representation, the anomalous dimensions are positive, and it seems like the family can be extended to spin zero to include the $X$ operator.[14] On the contrary, the $Y, Z, A,$ and $B$ representations show a comparably dense spectrum, which is consistent with the second scenario of a twist family below the double-twist operators. In Fig. 8 we display a cartoon that summarizes these observations; however, further numerical study will be needed to confirm or disprove this picture.

While it is encouraging that the spectrum plots are not inconsistent with the constraints from the lightcone bootstrap, we would like to address some issues that complicate the picture. As already mentioned, the uncertainty in the position of the kink induces uncertainty in the spectrum plots. Moreover, the plots should not be interpreted as displaying the full spectra in the respective representations since they are only showing operators with a non-negligible contribution to the $\phi$ four-point function. The complete spectrum of the theory is more dense than our cartoons of Fig. 8 indicate. For instance, the singlet representation is expected to contain accumulation points $2\tau_{\mathcal{O}}$ for all operators $\mathcal{O}$ in the theory and in our case this would include an accumulation point $2\Delta_Z = 1.2$, which is below the value $2\Delta_\phi$ in (40). A numerical bootstrap study of mixed correlators may reveal more operators. In [34], an ambitious attempt was made to determine the operator spectrum in the 3D Ising CFT, using a system of mixed correlators constructed out of $\sigma$ and $\epsilon$, the smallest dimension $\mathbb{Z}_2$ odd and even operators. Based on [45], the extremal functional was then applied to a sample of points on the boundary of the allowed region (in this case an island), and only stable operator dimensions were deemed to be candidates for primary operators in the spectrum. A similar computation was performed in [46] for the 3D $O(2)$ CFT. The method appears to give somewhat reliable predictions for the spectrum at low operator dimensions, but misses higher-spin operators asymptoting to double-twist dimensions of operators not included in the system of correlators studied.[15] It would be desirable to perform a similar study of the theory at hand.

---

[13]Note that in these theories, the anomalous dimensions never become particularly large; the largest values is attained in the 3D $O(2)$ CFT with $\tau_\infty - \tau_T = 0.0383$.

[14]By studying also the spectrum plots at spin 6 and 8 (not displayed), we note that the twists of the leading operator continues to decrease according to the behavior displayed in Fig. 8. Note however the feature at $\Delta = 5$ in Fig. 6f, a similar feature is noted at spin 6 (but not at spin 8) and is likely to be a spurious zero.

[15]For instance, results of [34] clearly identify the accumulation points $2\Delta_\sigma = 1.036$, $2\Delta_\epsilon = 2.825$ and

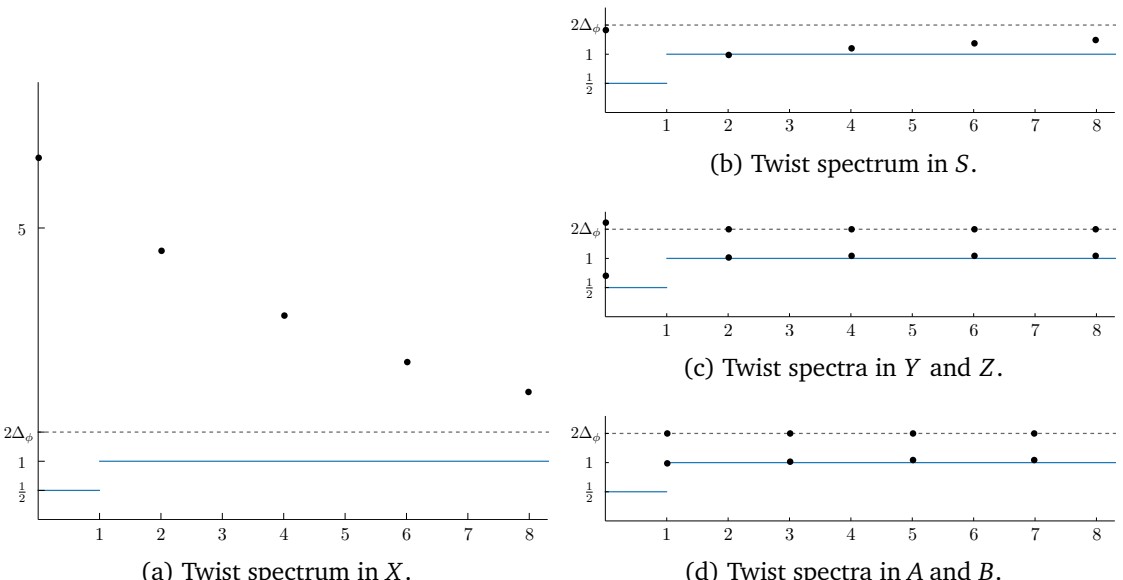

Figure 8: Cartoons for the hypothetical twist families in the various representations showing spin on the horizontal axis and twist on the vertical axis. This picture is based on the plots in Figs 6 and 7 using the extremal functional method, and needs to be confirmed or disproved by other methods. The unitarity bound (5) is shown in blue and the value $2\Delta_\phi = 1.5$ is shown in dashed gray. The complete spectrum is expected to contain more operators than those in the Regge trajectories shown. Only $S_2 = T$ and $A_1 = J$ have twists exactly on the unitarity bound.

## 5 Discussion

In this work we studied CFTs with global symmetry $MN_{m,n} = O(m)^n \rtimes S_n$ in $d = 3$ dimensions, with the motivation of evaluating further the potential existence of two distinct fixed points in such theories, as was recently suggested in [15] for the $MN_{2,2}$ and $MN_{2,3}$ theories. For theories with $n = 2$ we found evidence supporting this conclusion by considering various values of $m$ and observing two distinct kinks in bootstrap bounds, even for large $m$. For $n > 2$, our bootstrap bounds did not include a clear second kink in the expected region for large $n$, although such a kink does exist in the $MN_{2,3}$ theory (see [15, Fig. 4]). Our results suggest that there exists a critical line $n_c(m)$ below which there are two distinct MN CFTs.

The second kink we examined appears to not correspond to a known theory obtained within the standard Wilson–Fisher paradigm. Looking at the operator spectrum at this kink, we verified that it satisfies general expectations derived from the Nachtmann theorem [37] and the lightcone bootstrap [32, 33]. If this kink is due to a corresponding full-fledged CFT, this would indicate that the Wilson–Fisher paradigm is incomplete, i.e. that there exist fixed points in $d = 3$ that cannot be obtained from continuations of fixed points in $d = 4 - \varepsilon$. Recently, other bootstrap works have reported kinks that do not appear to be of Wilson–Fisher type [47–49], but the qualitative features of these kinks differ from ours and they may be of different origin. Interestingly, the picture emerging from our spectrum analysis at large $m$ shows some similarities with results in large $N$ $O(N)^3$ bosonic tensor models [50].[16] It would

---

$2\Delta_\sigma + 2 = 3.036$, but misses the intermediate values $2\tau_T = 2$ and $\tau_T + \Delta_\varepsilon = 2.413$. Likewise in the $O(2)$ CFT, not all expected operators in the charge 4 sector were found numerically in [46].

[16]In these models, $\Delta_\phi = d/4$, and the bilinear operators in some representations, including those containing

be interesting to further investigate the consistency of the spectrum of the second kink, using a numerical application of the Lorentzian inversion formula as has been done for the Ising and $O(2)$ CFTs [46, 53]; see also [34]. This would require a more precise determination of the spectrum and estimation of OPE coefficients.

With regards to the experimentally accessible $MN_{2,2}$ case, out of the five scenarios of [4, Sec. III.B.3], all of our bootstrap results favor "Scenario II", with two fixed points and therefore second-order phase transitions in both groups of systems. However, we stress that the values of the exponent $\beta$ obtained in experiments are in mild tension with the bootstrap ones for the first kink, see Tab. 2. Related to this issue is perhaps the fact that experimental and Monte Carlo results are inconsistent with unitarity, which predicts that $2\beta - \nu \geqslant 0$. Of course, by construction the numerical bootstrap results are consistent with unitarity. Given the relatively good agreement of $\nu$ between bootstrap, experimental and Monte Carlo results, we conclude that there is need for a more accurate determination of $\beta$ with experiments and Monte Carlo simulations, controlling systematic errors.

It would be desirable to compute results at large $m$ using conventional diagrammatic techniques similar to the large $N$ expansion of the critical $O(N)$ model. There, the leading scalar operator dimensions have been computed to orders $N^{-3}$ ($\Delta_\phi$ [54]) and $N^{-2}$ ($\Delta_S$ [55, 56] and $\Delta_T$ [57]), and results for the next-to-leading scalar singlet at order $N^{-2}$ also exist [58]. A difference with the $O(N)$ model is that the auxiliary field is now $X$, which is not a singlet under the global symmetry.

Additionally, it would be informative to perform a numerical bootstrap in intermediate dimensions $3 < d < 4$, like that in [59–61], in order to examine the behavior of the kinks as we approach $d = 4$. This will allow us to make better contact with the $\varepsilon$ expansion for the first kink, as well as determine if the second kink persists closer to $d = 4$. We note here that theories defined for non-integer values of $d$ are expected to be non-unitary [62], but this is not expected to cause problems when bounding scaling dimensions of low-lying operators. Mixed correlator bootstrap studies in $d = 3$ and $3 < d < 4$ could also yield crucial pieces of information that would allow us to further characterize the kinks. Finally, it would be instructive to apply alternative methods, such as non-perturbative RG, to MN theories; see [63] for successful recent work away from unitarity in $O(N)$ models.

# Acknowledgements

We would like to thank J. Gracey and A. Vichi for helpful discussions and comments on the manuscript. The project has received partial funding from the European Research Council (ERC) under the European Union's Horizon 2020 research and innovation programme (grant agreement no. 758903). This research used resources provided by the Los Alamos National Laboratory Institutional Computing Program, which is supported by the U.S. Department of Energy National Nuclear Security Administration under Contract No. 89233218CNA000001. Research presented in this article was supported by the Laboratory Directed Research and Development program of Los Alamos National Laboratory under project number 20180709PRD1.

---

conserved currents, acquire large anomalous dimensions, similar to Fig. 8a and 8b; see [51, 52] for recent work in the supersymmetric case.

## A  Large $m$ results for general $d$

In this appendix we present the results of section 3.1 for general spacetime dimension $d = 2\mu$. The scalar operator dimensions are given by

$$\Delta_\phi = \mu - 1 + \frac{n-1}{n}\frac{\eta_1}{m} + O(m^{-2}), \tag{41}$$

$$\Delta_S = 2(\mu - 1) + \frac{4(\mu-1)(2\mu-1)(n-1)}{(2-\mu)n}\frac{\eta_1}{m} + O(m^{-2}), \tag{42}$$

$$\Delta_X = 2 - 4\frac{(\mu-1)(2\mu-1)n - 4\mu^2 + 6\mu - 1}{(2-\mu)n}\frac{\eta_1}{m} + O(m^{-2}), \tag{43}$$

$$\Delta_Y = 2(\mu - 1) + \frac{4(n-1)}{(2-\mu)n}\frac{\eta_1}{m} + O(m^{-2}), \tag{44}$$

$$\Delta_Z = 2(\mu - 1) + 2\frac{(2-\mu)n - 2}{(2-\mu)n}\frac{\eta_1}{m} + O(m^{-2}), \tag{45}$$

where $\eta_1 = \frac{(\mu-2)\Gamma(2\mu-1)}{\Gamma(1-\mu)\Gamma(\mu)^2\Gamma(\mu+1)}$; the spinning operator dimensions by

$$\Delta_{S_\ell} = \ell + 2\Delta_\phi - \frac{2\mu(n-1)}{J^2 n}\left(\mu - 1 + \frac{\Gamma(\ell+1)\Gamma(2\mu-1)}{\Gamma(\ell+2\mu-3)}\right)\frac{\eta_1}{m} + O(m^{-2}), \tag{46}$$

$$\Delta_{X_\ell} = \ell + 2\Delta_\phi - \frac{2\mu}{J^2 n}\left((\mu-1)(n-1) + \frac{(n-2)\Gamma(\ell+1)\Gamma(2\mu-1)}{\Gamma(\ell+2\mu-3)}\right)\frac{\eta_1}{m} + O(m^{-2}), \tag{47}$$

$$\Delta_{Y_\ell} = \Delta_{A_\ell} = \ell + 2\Delta_\phi - \frac{2(\mu-1)\mu(n-1)}{J^2 n}\frac{\eta_1}{m} + O(m^{-2}), \tag{48}$$

$$\Delta_{Z_\ell} = \Delta_{B_\ell} = \ell + 2\Delta_\phi + \frac{2\mu(\mu-1)}{J^2 n}\frac{\eta_1}{m} + O(m^{-2}), \tag{49}$$

where $J^2 = (\mu - 1 + \ell)(\mu - 2 + \ell)$ and we have substituted the value for $a_X = \frac{(n-1)(\mu-1)\mu}{n(\mu-2)^2}\eta_1$; and the central charge corrections by

$$\frac{C_T}{C_{T,\text{free}}} = 1 - \frac{n-1}{(2-\mu)\mu(\mu+1)n}\left(2\mu(2-\mu)[\pi\cot(\pi\mu) + S_1(2\mu-2)] - \mu^2 + 2\mu + 4\right)\frac{\eta_1}{m}$$
$$+ O(m^{-2}), \tag{50}$$

$$\frac{C_J}{C_{J,\text{free}}} = 1 - \frac{2(2\mu-1)(n-1)}{\mu(\mu-1)n}\frac{\eta_1}{m} + O(m^{-2}), \tag{51}$$

where $S_1(x)$ denotes the standard analytic continuation of the harmonic numbers away from integer arguments. For $\mu = 3/2$ the expressions here reduce to those computed in section 3.1, and for $\mu = 2 - \varepsilon/2$, the expressions agree with the known results in the $\varepsilon$ expansion.

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
