# Peer review of "Perturbative and Nonperturbative Studies of CFTs with MN Global Symmetry"

_SciPost Physics, doi:SciPost Phys. 11, 015 (2021)_

## Round 1 · Referee Report · Anonymous (Referee 1) · 2021-4-14

Strengths
1- Very interesting method (bootstrap), which might provide an answer to long-standing controversies
Weaknesses
2 - Some confusion on the applications
Report
The paper discusses the so called mn model from the bootstrap perspective. The paper contains interesting physics, but the presentation requires some significant changes. I suggest publication after revision.
Requested changes
1) The model discussed here is usually called "mn model" in the literature, while the results for helimagnets and XY stacked triangular antiferromagnets are usually discussed using the "chiral" model. To avoid confusion, it is important to mention that for m=n=2 the two models can be mapped one onto the other. It should also be mentioned that none of the "epsilon"-expansion fixed points can describe such models (the only stable fixed point in the mn model cannot attract models in the chiral region) for such values of m and n.
2) After Eq. (1.2) the authors mention epsilon-expansion results. Within this approach, there are four fixed points. One can show nonperturbatively that the O(n) fixed point is stable (see A. Aharony,in Phase Transitions and Critical Phenomena, edited by C. Domb and J. Lebowitz Academic, New York, 1976, Vol. 6, p. 357). I think that a brief summary of the epsilon-expansion results would be needed, quoting the results of Aharony.
3) The authors report exponent estimates in Eq. (2.2) and then they compare them with "epsilon-expansion" results. I think the latter results should be reported. Second, it would be important to mention which fixed point is considered (if I'm note mistaken, it should be the mixed/cubic fixed point) among the four present.
4) Because of the mapping mn-model -> chiral model for m=n=2, in the literature there are also estimates of "non-epsilon-expansion" exponents, which are the object of the controversy mentioned at the beginning of p. 2. I think that the paragraph below Eq. (1.2) would be the proper place to present them. Are they close/very different from the results quoted in Eq. (1.2) ? It should be remarked that these
"non-epsilon-expansion" exponents are the only ones that can be compared with those for XY STAs.
5) The chiral model has been studied in two papers that I would suggest to quote:
Nakayama, Ohtsuki, Phys. Rev. D 89, 126009 (2014)
Nakayama, Ohtsuki, Phys. Rev. D 91, 021901(R) (2015)
Also in these two papers some "non-epsilon expansion" CFTs were found. Given that chiral and mn models agree for n=2,m=2, there should be a relation between the results obtained in the chiral setting and those obtained here. Some comments would be welcome. For instance, are the CFTs found for n=3,m=2 in the chiral model by Nakayama and Ohtsuki related (by smoothly changing n from 3 to 2) to the CFTs obtained here?
As an aside: is the approach of Nakayama, Ohtsuki (from a technical point of view) the same as the one used here? Do they implement the boostrap approach in the same way? Some comment is needed.
6) Fig. 1. Delta_phi and Delta_X are related to the standard critical exponents. It would be useful to report this relation in the caption. Probably, it would also help the reader to anticipate Eqs. (3.17) and (3.18) in the introduction .
7) The comparison made in Table 2 is not clear as it is mixing results of different nature. The experimental data refer to the chiral fixed point, and so do the MC results of Kawamura [I should mention that more accurate MC results are presented in Calabrese et al, Phys. Rev. B 70, 174439 (2004)]
On the other hand, the epsilon expansion results refer to the MN mixed point, which cannot describe the critical behavior of the XY STA. So they should not be compared with the STAs.
The bootstrap result quoted corresponds to kink 1. If I understand the analysis of the authors, it should not describe the STAs.
The only result result that might be relevant for the STAs is kink 2 which is not reported in the Table.
8) In the conclusions, the authors mention the failure of the epsilon-expansion. This is discussed at length by Calabrese et al. in the paper mentioned above. Another case is scalar electrodynamics: for N=2 there is a fixed point that cannot be obtained by using the epsilon expansion in the corresponding field theory [see Bonati et al., Phys. Rev. B 103, 085104 (2021)]. I think the discussion should be extended with more examples of systems where this phenomenon occurs.
Author: Johan Henriksson on 2021-06-03 [id 1486]
(in reply to Report 1 on 2021-04-14)
We thank the reviewer for their carefully written report and address the concerns in the list below.
1.After eq. (1.1) we have made it clear that there is a mapping between the $MN_{2,2}$ symmetry and the $O(2)^2\rtimes S_2$ symmetry, and that the fixed-point in the $\varepsilon$ expansion has $g<0$. Our non-perturbative methods are not able to access the details of the Lagrangian. Moreover, since we are not aware of any proof that the sign of the coupling at finite $\varepsilon$ agrees with that at infinitesimal $\varepsilon$, we do not wish to make any strong claims about the signs of the couplings in the three-dimensional theory.
2, 3.We have added some clarifications and a reference to the book chapter by Aharony. Note also that we give more details on the $\varepsilon$-expansion results in section 2.2.
4, 5.The purpose of the present paper is to study the large $m$ limit of $MN_{m,n}$ symmetric theories, and the main interest is not the case $m=n=2$. Indeed, in previous works, namely [1904.00017] and [2004.14388], the potential connection to the literature are discussed extensively.
6.The specific definition of $\Delta_X$ cannot be made precise until section~2, and while certainly one can define a critical exponent $\phi_X=\nu(d-\Delta_X)$ it is not among the standard critical exponents. We have decided to keep quations (3.17) and (3.18) in section 3 to avoid cluttering the introduction.
7.We wish to remain agnostic about identifying the results found using different methods, but have inserted a dashed line to make it clear that we do not claim that XY STAs, Tb and MC belong to the same fixed-point as found by the bootstrap. The table focuses on results for kink 1, and therefore we have not included estimates from kink 2. The XY STAs have been included since they appear to be closer to kink 1 than to kink 2.
8.We are discussing the paper by Calabrese et al.\ in our introduction. We do not agree with the reviewer that our situation is similar to Bonati et al. In that paper, the fixed-point found in the $\varepsilon$ expansion and the one expected for $N=2$ in 3d are have different symmetry groups. In our case we expect the same symmetry group.
Author: Johan Henriksson on 2021-06-03 [id 1487]
(in reply to Report 2 on 2021-05-11)We thank the reviewer for their carefully written report and address the points raised in the list below.
1.We have clarified that the relation kink-CFT is based on examples. There are some heuristic arguments for why this can be expected, but we do not want to dwell further upon this point.
2, 3.Clarifications added.
4.Equations (3.5)-(3.8) are exact in spin, and nothing in these equations indicate the contrary. Indeed, these results correspond to computing the perturbative inversion integral over the whole range $\bar z\in[0,1]$, as discussed in references [29,30].
5.We added the requested information in the caption of Fig. 4. In the introduction (first paragraph on p. 5) we clarified that by strength of numerics we mean the number of derivatives in the functional used in the numerical bootstrap computations. This is the standard terminology used in the bootstrap community.
6.This typo has been corrected, we thank the reviewer for pointing this out.
7.The determination of the value in eq. (4.1) was done by a combined visual analysis of figures 3 and 4. As $m$ increases, the position of the kink moves to the right, but as the numerical precision (number of derivatives) increases the position moves to the left. We have assigned a conservative error bar corresponding to the width of the graph in figure 4. Likewise, in estimating (4.2) and (4.3) the error bars assigned correspond to the size of the graph in figure 4 and are conservative. The value for the central point in (4.2) is truncated; had we displayed an additional significant figure, we agree with the reviewer that such value would be below $0.750$. A more precise determination of this central point would be sensitive to precisely how the position of the kinks is read off, and the precise extrapolation method to $\Lambda=\infty$ is performed. Given the instability of the position of the kink, we do not wish to enter into such a discussion.
8.We have added a comment to clarify this point.
9.We only read the spectrum and did not extract OPE coefficients. It would be interesting to pursue the referee's suggestion further.
10.Adding a gap $\Delta_{Y_4}>5.25$ does make the bound a little bit stronger, but does not change the qualitative picture with regards to the kink. Unfortunately this is not conclusive enough to prove or disprove the existence of an operator at spin $4$ near the unitarity bound in the $Y$ sector. We have added an additional comment that the schematic pictures in Fig. 8 needs to be confirmed or disproved by other methods and/or stronger numerics.

---

## Round 1 · Referee Report · Anonymous (Referee 2) · 2021-5-11

Report
The authors study conformal field theories in three dimensions with $MN_{m,n}= O(m)^n \rtimes S_n$ global symmetry by combining both numerical and analytical bootstrap techniques, and building on previous work by the authors. The author's work aims to shed light on an outstanding puzzle in the literature about the number of non-trivial fixed points for the case of $m=n=2$, where experimental data suggests the existence of two fixed points, while only one is found through epsilon expansions. The numerical bootstrap for systems with the given flavor symmetry display two "kinks", suggestive of the existence of two distinct CFTs for $n=2$ and any value of $m$. The authors study one of the kinks in a large $m$ expansion, using the inversion formula of Caron-Huot, in dimensions $2 < d \leq 4$. Near $d=4$ their results match those of the epsilon expansion connecting this kink to the fixed point accessible through the epsilon expansion. Their analytical results also match with numerical results at finite but large enough $m$. The authors then turn to the second kink, whose candidate CFT corresponds to a fixed point not visible in the epsilon expansion. The authors study the spectrum of this candidate CFT by determining position of the second kink with the numerical bootstrap and extracting the low lying spectrum for the case $m=100$. The obtained spectrum is shown to be consistent with Nachtmann's theorem and the light-cone bootstrap. Throughout the paper the authors compare their numerical and analytical results with those arising from experiments, epsilon expansion and Monte Carlo, and comment on some discrepancies between these results. The authors end listing several open directions for further study.
The paper is clearly written, with appropriate reviews of the essential background material, and contains new and interesting results. Therefore, I am happy to recommend the paper for publication after a few small changes and clarifications listed below:
Requested changes
1- In the introduction the authors state that a kink in the numerical plots "signals" the existence of a conformal field theory, perhaps the authors could clarify that this is a general expectation but that there is no guarantee that the particular CFT data near the kink corresponds to that of a fully consistent CFT, as they add later on.
2- In the introduction the authors refer to the "numerical precision of the computation" which presumably means the number of derivatives taken (and thus adding more physical constraints) - perhaps this can be clarified.
3- Also for the sake of the general reader, when above (2.4) is written that invariance under $x_1 \leftrightarrow x_2$ is satisfied by each conformal block it should be clarified that this follows from the fact that only even/odd spins appear in symmetric/antisymmetric representations as stated later about the leading twist operators.
4- Could the authors clarify if they computed the leading large spin contribution in the inversion formula (i.e. keeping only the $\bar{z} \to 1$ limit of t-channel blocks ) in which case eqs. (3.5) and following should have $+ \mathcal{O}(\ell^\#)$, or if they obtained the full $\bar{z}$ integral and have results valid for finite spin.
5- Could lambda be given for figure 3 as to make the comparison with fig. 4 easier? It is also perhaps worth explaining in a sentence in general terms what "numerical strength" means and how it relates to lambda.
6- In the sentence "The position for $\Delta_\phi$ of the kink is not stable" should it be $\Delta_X$?
7- Is the conclusion of eq. (4.1) from the plot of Fig. 3 with $m=1000$, or un-shown results? Relatedly, for $m=100$ estimating by eye from Fig. 4 the position of the kink with lambda=51 $\Delta_\phi$ seems lower than the value quoted in (4.2) - could the authors add a few words on how the position of the kink estimated?
8- In section 4.2 could the authors clarify what they mean that the theory does not approach a simplifying limit as $m \to \infty$, does this mean that the dimensions of most operators are not close to double-twist values?
9- When extracting the spectrum did the authors extract the OPE coefficients that solve the truncated crossing equations or only read the spectrum from the zeroes of the functional? In other words, could some of the operators have much smaller (or zero) OPE coefficients and perhaps even disappear in the full solution to crossing, or are they really expected to be present? This would be particularly relevant for the two families of almost conserved higher-spin currents.
10- Could the authors comment if there is there any confirmation of the existence of the almost-conserved twist family below the leading double-twist? E.g. if a gap in the spin $4$ Y spectrum is imposed such that a conserved current is excluded but not the spin four operator in the double-twist trajectory does the point of the kink in the $(\Delta_X,\,\Delta_\phi)$ plane become excluded?

---

## Round 2 · Referee Report · Anonymous (Referee 2) · 2021-6-18

Report

I thank the authors for their careful replies, and I appreciate the authors being conservative in the estimates even though the shown plots would suggest quite smaller error bars. The authors have addressed all my points, and I am happy to recommend the paper for publication. It can be published as is, although I have one minor final clarification to ask. Indeed eqs. (3.6-3.8) do not suggest a large spin expansion, the cause of confusion is the statement above (3.3) where it seems to say the dDisc is computed in the limit $v \ll 0$, and since the section is called large spin perturbation theory it was also not clear if non-perturbative finite spin effects were taken into account. (I am not sure [29,30] were what the authors meant to refer to in their reply.) For the paper to be self-contained, could the authors just briefly comment down to which spin they expect the inversion formula to give the correct answer, and thus their results to hold? In a generic CFT it would be for $\ell>1$, however the correlator being inverted here is an expansion for large $m$ - is the Regge growth expected to be the same for the $\mathcal{O}(m^{-1})$ piece of the correlator, or assumed to be? Or is it even better, since below (3.8) the results are evaluated for $\ell=1$? This issue is commented upon when discussing the $\ell=0$ evaluation of the authors' results but I believe it would be beneficial to comment a little bit earlier.

---

## Round 2 · Author Response

We thank both reviewers for their carefully written reports and their useful suggestions for improvements of the draft. We have made some changes following their reports.

---

## Round 2 · List of Changes

Introduction: Added some further clarifications and references (including the new footnote 3) concerning the literature discussion of non-perturbative fixed-points, and added some literature values for these. Clarified some formulations relating to the numerical bootstrap (regarding what a kink signals and numerical precision).

Table 2. Added a dashed line to separate experimental and MC data from kink 1 results. Added a new reference with additional MC result.

Figure 4. Added precision for qboot computations.

Section 4. Fixed typo. Clarified what we mean by a simplifying limit and that the cartoons in figure 8 need to be confirmed or disproved by further studies.

---

## Editorial Decision

published